# Prevalence of offering menopause hormone therapy among primary care doctors and its associated factors: A cross-sectional study

**Tiong Lim Low** [1]*, **Ai Theng Cheong** [2]*, **Navin Kumar Devaraj** [2], **Rohayah Ismail** [3]

**1** Klinik Kesihatan Jinjang, Kepong, Kuala Lumpur, Malaysia, **2** Department of Family Medicine, Faculty of Medicine and Health Sciences, Universiti Putra Malaysia, Serdang, Selangor, Malaysia, **3** Klinik Kesihatan Sentul, Kuala Lumpur, Malaysia

* cheaitheng@upm.edu.my (ATC); ivzh0120@gmail.com (TLL)

## Abstract

### Background

Guidelines recommend Menopausal Hormone Therapy (MHT) as the most effective treatment for menopausal symptoms. However, a local study found that the usage of MHT among menopausal women was low (8.1%), with one of the main reasons being it is not recommended by doctors. Therefore, the objectives of this study are to determine the prevalence of offering MHT in treating symptomatic menopausal women among primary care doctors (PCDs) and its associated factors.

### Methods

This cross-sectional study involved PCDs from the Federal Territory of Kuala Lumpur, the Federal Territory of Putrajaya and the state of Selangor. All PCDs provided services in government primary care clinics from the three states were invited through the doctor in charge of each clinic. An online survey links was provided for the participants to the self-administered questionnaire. The questionnaire included PCDs' demographics, their menopause management practices, attitudes towards MHT, perceived barriers in offering MHT, knowledge of related guidelines and received training on menopause management. The outcome variable was offering MHT which defined as either prescription of MHT or referral to hospital for MHT initiation. Multivariate logistic regression analysis was performed to determine the factors associated with offering MHT.

### Results

The response rate was 42.9% (559/1301). Of those who participated in the study, 77.8% of PCDs were female and 89.1% were medical officer. Although 66.9% of the participants reported offering MHT to their patients, the actual prescription rate was low (0.9%). Most PCDs (66%) would refer the patients to hospitals. 87.1% of PCDs (487/559) reported that MHT was not available in their clinic. In the past 12 months, 83% of PCDs had not received any related training. Female PCDs (AOR:2.5, CI: 1.51–4.13, p<0.001), perceiving MHT as

**Data Availability Statement:** The data contains potentially identifiable information about healthcare providers, including age, gender, position, place of practice, years of service, and menopause

experience within the study population. Contact information for a data access: Medical Research and Ethics Committee, National Institutes of Health, Ministry of Health Malaysia, Block A, Level 2, No 1, Jalan Setia Murni U13/52, Seksyen U13, Setia Alam, 40170, Shah Alam, Selangor, Malaysia Telephone: +603-3362 8398. mrecsec@moh.gov. my.

**Funding:** The author(s) received no specific funding for this work.

**Competing interests:** NO authors have competing interests.

preference treatment for menopause symptom (AOR:3.6, CI: 2.13–6.19, p < 0.001), having likelihood to recommend MHT to family and friends (AOR:3.0, CI: 1.87–4.83, p < 0.001), and receiving training on menopause management (AOR:2.7, CI: 1.30–5.56, p = 0.008) were the positive predictor of offering MHT. The negative predictors in offering MHT were no-experience in prescribing MHT (AOR: 0.4, CI: 0.15–0.87, p = 0.024) and lack of information regarding MHT for the patient (AOR: 0.4, CI:0.20–0.67, p < 0.001).

## Conclusion

The study revealed a low rate of MHT prescription among PCDs, with many relying on referrals to hospitals for managing menopausal symptoms. The findings underscore the need for strategies that includes fulfilling professional training gaps, improving MHT availability, and improving information dissemination for patient.

## Introduction

Primary care clinics often act as the first point of contact for management of menopausal symptoms. Primary care doctors (PCDs) play a crucial role in symptom management, risk assessment, and referral to specialized menopause services for complex cases. In Australia, primary care doctors serve as coordinators within dedicated menopause health hubs [1], while in Malaysia, the Health Screening Program (BSSK) entails primary care clinics to screen for menopausal symptoms [2]. Ideally, patients should discuss their symptoms with doctors to determine the most appropriate treatment course, most oftenly Menopause Hormone Therapy (MHT), barring contraindications [3, 4].

However, there was a decline in MHT prescription rates following the reporting of the results of Women's Health Initiative Study in 2002, primarily due to doctors' concerns about its risks of developing breast cancer [5, 6]. However recently, MHT usage has begun to increase. In United Kingdom, prescriptions of MHT have doubled in five years (2017–2022) due to media promotion and stakeholder engagement [7]. Conversely, in Malaysia, MHT usage among menopausal women remains low (8.1%), one of the main reasons is because doctors do not frequently recommend it [8].

PCDs, serving as health service gatekeepers, are vital for educating and increasing awareness about menopause and its treatments. However, no study in Malaysia so far has explored PCDs' strategies for managing symptomatic menopausal women. As a lack of doctor recommendation significantly contributes to Malaysia's low MHT usage, it is essential to understand the factors influencing PCDs' practices in managing menopause.

## Materials and methods

This cross-sectional study was conducted from 1st June to 31st October 2022, utilizing universal sampling. The study population comprised PCDs currently practicing in public health clinics in the state of Selangor and the Federal Territories of Kuala Lumpur and Putrajaya. Inclusion criteria were PCDs that have registered with the Malaysia Medical Council and possessing a valid Annual Practice Certificate with at least 12 months of experience practising in primary care clinics. PCDs who did not treat patients at least once a week were excluded.

Sample size was calculated based on two population proportions hypothesis testing and referring to information available in literature, i.e. the study by Nilsen et al., that showed a gender difference in MHT prescription rates: 74% for female doctors as compared to 63% for male

doctors [9]. With the power of study of 80% and significance level, α at 0.05 and 95% confidence interval, the sample size was determined to be 558 but considering a typical online survey response rate of about 45% [10], we approached all 1301 eligible PCDs from the study sites.

Ethical approval for this study was obtained from the Medical Research and Ethics Committee (MREC), Ministry of Health Malaysia (NMRR ID-22-00581-UD6). Prior to participant recruitment, approval to conduct the study was also obtained from the State Health Department of Selangor and the Federal Territories of Kuala Lumpur and Putrajaya, as well as the relevant district health departments. List of primary health clinic (22 clinics under Kuala Lumpur and Putrajaya, 81 clinics under Selangor) was obtained from Unit Primer under the state health offices. The contacts of the doctors in charge of the clinics were obtained through communication with the district health officers. The numbers of PCDs who work in the respective clinic was obtained through the doctor in charge of the clinic, of which the total number of potentially eligible PCDs were found to be 1301 doctors. For recruitment of participants in the study, an advertisement of invitation for participating in the survey, the approval letter from the state and the link for online survey (Google Form) were sent through WhatsApps to doctor in charge of each clinic. The doctor in charge of each clinic then distributed the online survey form to all PCDs in their clinic. The doctors in charge were given reminders twice a month (up to 6 times during the survey period) to circulate the survey form to the respondents.

The tool used in this study is a questionnaire originally developed by previous researchers to assess health professionals' attitudes regarding menopause hormone therapy in Australia [11]. The questionnaire is in English and permission was obtained from the original authors to use it in our study. To better suit our local context, we made some adaptations to the original questionnaire. The questionnaire items were analyzed individually as categorical data, and no composite scoring was used. Content validation of the adapted questionnaire was performed by three senior family medicine specialists. Additionally, a pilot study was conducted involving 30 health professionals, who were excluded from the final sample. Feedback from the pilot study led to minor corrections, primarily in grammar and wording, to enhance clarity and ensure the questions were clearly understood by the respondents.

The first part of the questionnaire consisted of the information about the study and consent. Respondent required to check "Agree" on the consent to proceed with the online survey. Other domain of the questionnaire in the study consists of PCDs' demographics, their menopause management practices, attitudes towards MHT, perceived barriers in offering MHT, knowledge of related guidelines and whether they received training on menopause management (refer S1 File).

## Data analysis

Data analysis was conducted using SPSS version 28. The outcome variable was the practice of offering MHT; this included the prescription of MHT or referrals to hospitals for MHT initiation. The factors examined were the PCDs' socio-demographic characteristic, clinic profile for menopausal care, PCDs' attitude towards MHT, PCDs' awareness of clinical practice guidelines, PCDs' ability to keep up with evidence on menopause management and whether the PCDs' received training on menopause management in 12 months and PCDs' perceived barrier in offering MHT.

Categorical data were presented as frequencies and percentages. A simple and multivariate logistic regression analysis was conducted to determine the factors that would influence the outcome of offering MHT. The factors with p value <0.25 in simple logistic regression analysis were included in the multivariate model [12]. Variance Inflation Factor (VIF) was examined to determine the degree of multicollinearity between the factors. Enter method

was used for variable selection in multivariable analysis. The Hosmer-Lemeshow test was performed to assess the model's goodness of fit, and the discriminative/predictive power was evaluated by Receiver Operating Characteristic (ROC) curve analysis. A p-value > 0.05 in the Hosmer-Lemeshow statistics considered that the model fits the data. Then, the regression model was assessed to determine whether it fits the observed data well by examining the residuals. The residuals examined were Standardised residuals and Studentised residuals. Influence statistics such as Cook's distance, DFBeta, and leverage were also used. The regression model fits the observed data well if the value for Cook's distance is less than 1, the leverage value lies between 0 and 1, the Studentized and standardised residual is less than 3, and DFBeta is less than 1.

A significance level of < 0.05 was applied to all statistical tests, and data were presented with a 95% confidence interval.

## Result

This study received a response rate of 42.9% (559/1301). The majority of respondents were female (77.8%) and held the position of medical officers (89.1%). A high portion of respondents, 77.1%, reported encountering at least one patient presenting with menopausal symptoms per month (Refer Table 1). For the management of patients with menopausal symptoms, only 0.9% (5/559) of PCDs actively prescribed MHT, with 66% (369/559) referring their patients to hospital for MHT initiation. 87.1% PCD reported MHT was not available in their clinic. Regarding PCDs' attitudes towards MHT, only 36% of respondents considered MHT as a preferred treatment option for management of menopausal symptoms. About 60% of the PCDs were inclined to recommend MHT to family and friends, but less than half of them had a predilection for personal usage (Refer Table 1). In terms of awareness of the clinical practice guidelines (CPG) on menopause management, 39.7% of PCDs were not aware of its existence. Moreover, 39.2% reported difficulties in staying up to date with the current evidence of management, and 83% reported not having received any training on the management of menopause in the past 12 months (Refer Table 1).

For the barriers in offering MHT, there were a high proportion of PCDs that reported a lack of experience in prescribing MHT (90.2%), limited availability of MHT (85.2%), and concerns of patients about the risks of breast cancer associated with MHT (79.8%) (Refer Table 2).

The multivariate logistic regression model, detailed in Table 3, was used to examine factors associated with PCDs' practice of offering MHT. Regarding sociodemographic characteristics, only gender played a significant role in the decision to offer MHT. Female PCDs were 2.5 times more likely to offer MHT than their male counterparts (AOR:2.5, 955 CI: 1.51–4.13, p<0.001).

A key factor influencing the decision to offer MHT was the PCDs' attitude towards the treatment. Those PCDs who viewed MHT as the preferred treatment for menopause symptoms were 3.6 times more likely to offer it (AOR:3.6, CI: 2.13–6.19, p < 0.001). Similarly, PCDs who were likely to recommend MHT to family and friends were three times more likely to offer the treatment (AOR:3.0, CI: 1.87–4.83, p < 0.001). In addition, PCDs who had received training on menopause management in the past 12 months were 2.7 times more likely to offer MHT (AOR:2.7, CI: 1.30–5.56, p = 0.008). In terms of perceived barriers, two significant variables were associated with practice to offer MHT. PCDs with no experience prescribing MHT (AOR: 0.4, CI: 0.15–0.87, p = 0.024) and those citing a lack of patient information about MHT (AOR: 0.4, CI:0.20–0.67, p < 0.001) were less likely to offer the treatment.

**Table 1. PCD's Socio-demographic characteristics, clinic profile for menopausal care, attitude towards MHT, professional training on menopause management (N = 559).**

| Variables | | Frequency | Percentage |
|---|---|---|---|
| | | N = 559 | (%) |
| **PCD's Socio-demographic characteristics** | | | |
| Age, years | <35-year-old | 154 | 27.5 |
| | 35–39-year-old | 267 | 47.8 |
| | ≥40-year-old | 138 | 24.7 |
| Gender | Male | 124 | 22.2 |
| | Female | 435 | 77.8 |
| Ethnic | Malay | 316 | 56.5 |
| | Chinese | 108 | 19.3 |
| | Indian | 113 | 20.2 |
| | Other | 22 | 4.0 |
| Position | Medical officer | 498 | 89.1 |
| | Family medicine specialist | 61 | 10.9 |
| Year of practice in primary care, years | <5 years | 169 | 30.2 |
| | 5-9years | 209 | 37.4 |
| | ≥10 years | 181 | 32.4 |
| Personal or family member experienced with menopausal symptom | Yes | 55 | 9.8 |
| | No | 504 | 90.2 |
| **Clinic profile for menopausal care** | | | |
| Number of encounters of patients with symptomatic menopause in past 1 month | None | 128 | 22.9 |
| | 1–2 | 338 | 60.5 |
| | >2 | 93 | 16.6 |
| Available of MHT in practice | Yes | 72 | 12.9 |
| | No | 487 | 87.1 |
| **PCD's attitude toward MHT** | | | |
| Preference treatment for menopause symptom | MHT | 201 | 36.0 |
| | Non-MHT | 358 | 64.0 |
| Likelihood of recommending MHT to family and friends | Likely | 348 | 62.3 |
| | Neutral | 162 | 29.0 |
| | Unlikely | 49 | 8.8 |
| Likelihood of self-usage of MHT* | Likely | 199 | 45.7 |
| | Neutral | 160 | 36.8 |
| | Unlikely | 76 | 17.5 |
| **PCD's professional training on menopause management** | | | |
| Awareness of CPG—management of menopause in Malaysia | Yes | 337 | 60.3 |
| | No | 222 | 39.7 |
| Ability to keep up with current recommendations/ guidelines/ evidence on menopause management | Yes | 67 | 12.0 |
| | Moderately | 273 | 48.8 |
| | No | 219 | 39.2 |
| Training received on menopause management in the past 12 months | Attachment in menopause clinic | 2 | 0.4 |
| | Twice or more | 5 | 0.9 |
| | Once | 88 | 15.7 |
| | None | 464 | 83.0 |

MHT: Menopause Hormone Therapy

PCD: Primary care Doctor

CPG: Clinical Practical Guideline

*For Female Respondent only

Table 2. The PCD's perceived barrier in offering MHT (N = 559).

| The PCDs' perceived barrier in offering MHT | Frequency | Percentage |
|---|---|---|
| | N = 559 | (%) |
| No experience in prescribing MHT | 504 | 90.2 |
| MHT is not widely available | 476 | 85.2 |
| Patients' concerns about the risk of breast cancer with MHT | 446 | 79.8 |
| Lack of information regarding MHT for the patient | 440 | 78.7 |
| Time constraints when discussing MHT | 434 | 77.6 |
| PCDs' concerns regarding side effects of MHT | 376 | 67.3 |
| Patients' concerns about other non-breast cancer risks with MHT | 355 | 63.5 |
| MHT is expensive | 286 | 51.2 |
| Patient preference for complementary/ alternative therapies | 245 | 43.8 |

## Discussion

### Principal of findings

A total of 66.9% of the participants reported offering MHT to their patients. However, the actual prescription rate was low (0.9%). Most PCDs (66%) would refer the patients to hospitals. The positive predictor of offering MHT include female PCDs, perceiving MHT as preference treatment for menopause symptom, having likelihood to recommend MHT to family and friends, and receiving training on menopause management. The negative predictors in offering MHT were no-experience in prescribing MHT and lack of information regarding MHT for the patient.

### Comparison with literature

Most PCDs in our study treated more than one patient per month with menopausal symptoms. Despite this, only about 1% of PCDs prescribed MHT to their patients. This rate is significantly low compared to the 7% prescription rate in a United Kingdom primary care study in 2022 [13]. Instead, patients were often referred to hospital for MHT initiation. This practice could result in problems such as discontinuity of care, potential loss of patient follow-ups in primary care, increase hospital patients load resulting in longer hospital appointment and waiting times. Ultimately, this process might delay patients from getting the treatment and lead to prolonged patient suffering from menopausal symptoms and affecting their quality of life [14].

Our research indicated that female PCDs were more likely to offer MHT than their male counterparts. Findings from other studies have shown variance. For example, a Spanish study reported male physicians were more inclined to prescribe MHT [15], whereas studies from the United States found that female doctors were more likely to do so [16, 17]. We hypothesize that our finding could stem from female PCDs demonstrating more empathy towards women's health-related issues [18].

The attitude of PCDs towards MHT had a significant influence on their likelihood to offer this treatment. In our study, PCDs who considered MHT as the preferred treatment for menopause symptoms and those who would recommend MHT to their family and friends were more likely to offer MHT to their patients. However, compared to overseas studies, our study reported a less favourable attitude towards MHT. For instance, only 36% of PCDs in our study agreed that MHT is the preferred treatment for menopausal symptoms, a stark contrast to the 87% to 91% of health professionals in Australia and China [11, 19]. Despite a more favourable attitude towards MHT among Australian health professionals, recent studies indicate that primary care services in Australia still struggle to treat their patients with menopause effectively

**Table 3. Logistic regression model on factor associatede with practice to offer MHT in treating symptomatic menopause women (N = 559).**

| Determinants[β] | Not Offer MHT | Offer MHT | Simple Logistic Regression | | Multivariate Logistic Regression | |
|---|---|---|---|---|---|---|
| | n (%) | n (%) | Crude OR | p-value | Adjusted OR | p-value |
| | | | (95% CI) | | (95% CI) | |
| **PCDs' sociodemographic characteristics** | | | | | | |
| Age, years | | | | | | |
| <35-year-old | 49 (26.5) | 105 (28.1) | 1.0 | 0.776 | | |
| 35–39-year-old | 87 (47.0) | 180 (48.1) | 0.9 (0.63–1.47) | 0.505 | | |
| ≥40-year-old | 49 (26.5) | 89 (23.8) | 0.8 (0.521–1.37) | 0.555 | | |
| Gender | | | | | | |
| Male | 61 (33.0) | 63 (16.8) | 1.0 | | 1.0 | |
| Female | 124 (67.0) | 311 (83.2) | 2.4 (1.61–3.65) | <0.001[α] | 2.5 (1.51–4.13) | <0.001* |
| Race | | | | | | |
| Malay | 95 (51.4) | 221 (59.1) | 1.0 | | | |
| Chinese | 39 (21.1) | 69 (18.4) | 0.7 (0.48–1.20) | 0.862 | | |
| Indian | 44 (23.8) | 69 (18.4) | 0.6 (0.43–1.05) | 0.701 | | |
| Others | 7 (3.8) | 15 (4.0) | 0.9 (0.36–2.33) | 0.530 | | |
| Year of practice in primary care, years | | | | | | |
| <5 years | 62 (33.5) | 107 (28.6) | 1.0 | 0.163[α] | 1.00 | 0.269 |
| 5-9years | 59 (31.9) | 150 (40.1) | 1.5 (0.95–2.27) | 0.796 | 1.4 (0.86–2.46) | 0.158 |
| ≥10 years | 64 (34.6) | 117 (31.3) | 1.1 (0.68–1.63) | 0.131[α] | 1.0 (0.57–1.76) | 0.971 |
| Position | | | | | | |
| Medical officer | 171 (92.4) | 327 (87.4) | 1.0 | | 1.0 | |
| FMS | 14 (7.6) | 47 (12.6) | 1.8 (0.94–3.27) | 0.077[α] | 1.4 (0.65–3.25) | 0.359 |
| Personal/Family experience with menopausal symptom | | | | | | |
| No | 178 (96.2) | 326 (87.2) | 1.0 | | 1.0 | |
| Yes | 7 (3.8) | 48 (12.8) | 3.7 (1.65–8.44) | 0.001[α] | 1.9 (0.72–4.89) | 0.191 |
| **Clinical profile for menopausal care** | | | | | | |
| Number of encounters of patient with symptomatic menopause in past 1 month | | | | | | |
| None | 37 (20.0) | 91 (24.3) | 1.0 | 0.455 | 1.0 | 0.407 |
| 1–2 | 114 (61.6) | 224 (59.9) | 0.8 (0.51–1.24) | 0.230[α] | 0.7 (0.40–1.18) | 0.180 |
| >2 | 34 (18.4) | 59 (15.8) | 0.7 (0.39–1.24) | 0.611 | 0.8 (0.370–1.52) | 0.426 |
| Available of MHT in practice | | | | | | |
| No | 171 (92.4) | 316 (84.5) | 1.0 | | 1.0 | |
| Yes | 14 (7.6) | 58 (15.5) | 2.2 (1.21–4.13) | 0.010[α] | 1.5 (0.74–3.07) | 0.251 |
| **PCD's attitude towards MHT** | | | | | | |
| MHT as the preferred treatment for menopause symptom | | | | | | |
| No | 159 (85.9) | 199 (53.2) | 1.0 | | 1.0 | |
| Yes | 26 (14.1) | 175 (46.8) | 5.4 (3.38–8.53) | <0.001[α] | 3.6 (2.13–6.19) | <0.001* |
| No | 159 (85.9) | 199 (53.2) | 1.0 | | 1.0 | |
| Yes | 26 (14.1) | 175 (46.8) | 5.4 (3.38–8.53) | <0.001[α] | 3.6 (2.13–6.19) | <0.001* |
| Likelihood of self-usage | | | | | | |
| Unlikely | 52 (28.1) | 49 (13.1) | | | | |
| Neutral | 86 (46.5) | 124 (33.2) | | | | |
| Likely | 47 (25.4) | 201 (53.7) | 2.2 (1.72–2.81) | <0.001[α] | 1.0 (0.64–1.51) | 0.948 |
| Likelihood of recommending MHT to family and friends | | | | | | |
| Unlikely | 33 (17.8) | 16 (4.3) | | | | |
| Neutral | 84 (45.5) | 78 (20.9) | | | | |
| Likely | 68 (36.8) | 280 (74.9) | 3.4 (2.54–4.58) | <0.001[α] | 3.0 (1.87–4.83) | <0.001* |

*(Continued)*

**Table 3.** (Continued)

| Determinants[β] | Not Offer MHT | Offer MHT | Simple Logistic Regression | | Multivariate Logistic Regression | |
|---|---|---|---|---|---|---|
| | n (%) | n (%) | Crude OR | p-value | Adjusted OR | p-value |
| | | | (95% CI) | | (95% CI) | |
| **PCD's professional training in menopausal management** | | | | | | |
| *Awareness of CPG in the management of menopause* | | | | | | |
| No | 91 (49.2) | 131 (35.0) | 1.0 | | | |
| Yes | 94 (50.8) | 243 (65.0) | 1.8 (1.25–2.56) | <0.001[α] | 1.0 (0.64–1.64) | 0.902 |
| *Ability to keep up with evidence on menopause management* | | | | | | |
| No | 93 (50.3) | 126 (33.7) | | | | |
| Moderately | 76 (41.1) | 197 (52.7) | | | | |
| Yes | 16 (8.6) | 51 (13.6) | 1.7 (1.26–2.20) | <0.001[α] | 0.9 (0.62–1.29) | 0.573 |
| *Training received on menopause management in the past 12 months* | | | | | | |
| None | 172 (93.0) | 292 (78.1) | | | | |
| Once | 13 (7.0) | 75 (20.1) | | | | |
| Twice or more | 0 (0.0) | 5 (1.3) | | | | |
| Attachment in menopause clinic | 0 (0.0) | 2 (0.5) | 3.6 (1.96–6.50) | <0.001[α] | 2.7 (1.30–5.56) | 0.008* |
| **PCD's Perceived Barrier to offer MHT** | | | | | | |
| *No Experience in prescribing MHT* | | | | | | |
| No | 12 (6.5) | 43 (11.5) | 1.0 | | 1.0 | |
| Yes | 173 (93.5) | 331 (88.5) | 0.5 (0.27–1.04) | 0.065[α] | 0.4 (0.15–0.87) | 0.024* |
| *MHT is not widely available* | | | | | | |
| No | 28 (15.1) | 55 (14.7) | 1.0 | | | |
| Yes | 157 (84.9) | 319 (85.3) | 1.0 (0.63–1.69) | 0.893 | | |
| *Patient concerned about the risk of breast cancer with MHT* | | | | | | |
| No | 35 (18.9) | 78 (20.9) | 1.0 | | 1.0 | |
| Yes | 150 (81.1) | 296 (79.1) | 0.50 (0.32–0.76) | 0.001[α] | 0.50 (0.28–1.04) | 0.067 |
| *Lack of information regarding MHT by the patient* | | | | | | |
| No | 15 (8.1) | 104 (27.8) | 1.0 | | 1.0 | |
| Yes | 170 (91.9) | 270 (72.2) | 0.50 (0.32–0.73) | <0.001[α] | 0.4 (0.20–0.67) | <0.001* |
| *Time constraints when discussing MHT* | | | | | | |
| No | 41 (22.2) | 84 (22.5) | 1.0 | | | |
| Yes | 144 (77.8) | 290 (77.5) | 1.0 (0.66–1.55) | 0.937 | | |
| *PCDs' concerns regarding side effects of MHT* | | | | | | |
| No | 58 (31.4) | 125 (33.4) | 1.0 | | | |
| Yes | 127 (68.6) | 249 (66.6) | 1.1 (0.75–1.60) | 0.623 | | |
| *Patient Concerned about Other Non-Breast Cancer risks with MHT* | | | | | | |
| No | 65 (35.1) | 145 (38.8) | 1.0 | | 1.0 | |
| Yes | 120 (64.9) | 229 (61.2) | 0.60 (0.39–0.81) | 0.002[α] | 0.8 (0.42–1.38) | 0.371 |
| *Patient preferences for complementary/ alternative therapies* | | | | | | |
| No | 90 (48.6) | 90 (48.6) | 1.0 | | 1.0 | |
| Yes | 95 (51.4) | 95 (51.4) | 0.6 (0.44–0.90) | 0.12[α] | 0.9 (0.56–1.54) | 0.789 |
| *MHT is expensive* | | | | | | |
| No | 86 (46.5) | 187 (50.0) | 1.0 | | | |

(*Continued*)

**Table 3.** (Continued)

| Determinants[β] | Not Offer MHT | Offer MHT | Simple Logistic Regression | | Multivariate Logistic Regression | |
|---|---|---|---|---|---|---|
| | n (%) | n (%) | Crude OR | p-value | Adjusted OR | p-value |
| | | | (95% CI) | | (95% CI) | |
| Yes | 99 (53.5) | 187 (50.0) | 1.2 (0.80–1.63) | 0.434 | | |

[β]Offer MHT—Refer patient to hospital for MHT initiation or Prescribe MHT.

*P-value <0.05 in multivariate logistic regression

[α] variable with P-value<0.25 in univariate simple logistic regression were included in the final multivariate logistic regression model

Hosmer Lemeshow test (p>0.05).

Variance Inflation Factor ranged 1.05 to 2.25, standardized residuals ranged -2.78 to 2.16, Studentized residuals ranged -2.41 to 2.22. Cook's distance ranged 0.001 to 0.66, DFBeta ranged -1.43 to 0.66, leverage ranged 0.004 to 0.155

[20, 21]. This suggests that even with a positive perception of MHT, challenges in implementing comprehensive menopause care persisted.

We found that the availability of menopausal hormone therapy (MHT) and experience in prescribing it were significant factors in offering MHT. Despite MHT being listed in the Ministry of Health formulary for primary care clinics [22], its availability was low in most practices. This lack of access limits PCDs opportunities to learn how to use MHT, discuss the treatment with patients, and confidently prescribed it.

Regarding professional training, a notable 83% of PCDs reported not having received any related training in the past 12 months on menopause management. Approximately 40% were unaware of the CPG and found it challenging to stay up to date with the latest recommendations on menopause management. This situation mirrored those in other countries, where gaps existed in knowledge specifically about menopause management and the use of MHT [21, 23, 24]. Healthcare providers are often knowledgeable about menopause itself but are uncertain about its management and treatment.

Thus, providing regular training and refresher courses on menopausal management would be helpful in strengthening the knowledge of PCDs and improving overall menopause care.

Another significant barrier identified was the lack of information about menopausal hormone therapy (MHT) available for patients, with 78.7% of primary care doctors (PCDs) perceiving this as an obstacle in this study. Studies in Malaysia have shown that most patients obtain information about menopause treatment from friends, family, and media, and they often prefer complementary and alternative medicine, which may not be evidence-based [8, 25]. This situation poses additional challenges for PCDs, as they must usually address these misconceptions within limited consultation time [21]. Conversely, in United Kingdom, due to increased awareness about menopause management among public with more dependable information about MHT, there is an increased health seeking behaviour among public for the menopausal symptoms, leading to an increase in MHT prescription [7]. Therefore, improving the dissemination of accurate and evidence-based information about MHT is essential to enhance patient care and treatment outcomes.

## Strength and limitations

The strength of this study lies in its pioneering nature, as it is among the first studies conducted in Malaysia to understand the prevalence of offering MHT for managing symptomatic menopausal women among PCDs and the associated factors.

There were some limitations in this study. Firstly, the study was conducted in public health clinics and thus the outcomes results might not represent practices in private clinics, which

also plays a significant role in the primary care setting in the country. Secondly, the study's outcomes, which focus on the practice of offering MHT, rely on respondents recalling their practices over the past 12 months. This could introduce recall bias, potentially leading to under-reporting or over-reporting of the results.

## Conclusion

The study revealed a low rate of MHT prescription among PCDs, with many relying on referrals to hospital for managing menopausal symptoms. The findings underscore the need for strategies that includes fulfilling professional training gaps, improving MHT availability, and improving information dissemination for patient. These improvements are essential for enhancing the menopausal care to patient.

## Supporting information

**S1 File. Questionnaire for data collection.**
(DOCX)

## Acknowledgments

We would like to thank the Director-General of Health Malaysia for his permission to publish this article. We would like to extend our appreciation to all respondents of the study.

## Author Contributions

**Conceptualization:** Tiong Lim Low, Ai Theng Cheong, Navin Kumar Devaraj, Rohayah Ismail.

**Data curation:** Tiong Lim Low.

**Formal analysis:** Tiong Lim Low.

**Investigation:** Tiong Lim Low.

**Methodology:** Tiong Lim Low, Ai Theng Cheong, Navin Kumar Devaraj.

**Project administration:** Tiong Lim Low.

**Resources:** Tiong Lim Low.

**Software:** Tiong Lim Low.

**Supervision:** Ai Theng Cheong, Navin Kumar Devaraj, Rohayah Ismail.

**Validation:** Tiong Lim Low, Ai Theng Cheong.

**Visualization:** Tiong Lim Low.

**Writing – original draft:** Tiong Lim Low.

**Writing – review & editing:** Ai Theng Cheong, Navin Kumar Devaraj, Rohayah Ismail.

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
