## [Decision Letter · Decision Letter 0]

31 Jul 2024

PONE-D-24-25801Prevalence of offering menopause hormone therapy among primary care doctors and its associated factors: A cross-sectional studyPLOS ONE

Dear Dr. Cheong,

Thank you for submitting your manuscript to PLOS ONE. After careful consideration, we feel that it has merit but does not fully meet PLOS ONE’s publication criteria as it currently stands. Therefore, we invite you to submit a revised version of the manuscript that addresses the points raised during the review process.

We look forward to receiving your revised manuscript.

Kind regards,

Ghanshyam G. Tejani

Academic Editor

PLOS ONE

Journal Requirements:

**Additional Editor Comments:**

Thank you for submitting your manuscript. The reviewers have suggested minor revisions. Please revise and resubmit your manuscript based on their suggestions.

Reviewers' comments:

Reviewer's Responses to Questions

**Comments to the Author**

1. Is the manuscript technically sound, and do the data support the conclusions?

Reviewer #1: Yes

Reviewer #2: Yes

2. Has the statistical analysis been performed appropriately and rigorously? 

Reviewer #1: Yes

Reviewer #2: Yes

3. Have the authors made all data underlying the findings in their manuscript fully available?

Reviewer #1: Yes

Reviewer #2: Yes

4. Is the manuscript presented in an intelligible fashion and written in standard English?

Reviewer #1: Yes

Reviewer #2: Yes

5. Review Comments to the Author

**Reviewer #1: **Well done to the authors in doing this important study.

In the method, suggest to include brief input of the outcome on the content and face validity of the questionnaire if there were any issues?

Please standardize the term use

ie the doctors in-charge or the in-charge doctors?

p value less than (to use symbol <)

**Reviewer #2:** Overall it's a very good study and very relevant to primary care. As a pioneer study, i think subsequent study can explore the reason for low prescription of MHT in relation to patients perception towards menopause symptoms and MHT. I just want clarification regarding the questionnaire used. in the manuscript its written adapted from previous study and given the reference number (11). But in the references, reference number 11 is about 'multiple logistic regression. ( 11. Jr D, Lemeshow S, Sturdivant R. The Multiple Logistic Regression Model. In 2013. p. 35–47. ) I can't find the reference to the questionnaire used in this study. And what step they do... is it only content validation? what language the original questionnaire used. Please provide me the original one and the method you use to validate the questionnaire.

6. PLOS authors have the option to publish the peer review history of their article (what does this mean?). If published, this will include your full peer review and any attached files.

Reviewer #1: No

Reviewer #2: **Yes: **Nurul Husna Binti Azmi

---

## [Author Response · Author response to Decision Letter 0]

21 Aug 2024

Reviewer #1: Well done to the authors in doing this important study.

In the method, suggest to include brief input of the outcome on the content and face validity of the questionnaire if there were any issues?

Thank you to the reviewer for acknowledge this study.

We have added more details for clarity. (Page 6, line 8 to line 10)

' Feedback from the pilot study led to minor corrections, primarily in grammar and wording, to enhance clarity and ensure the questions were clearly understood by the respondents.'

Please standardize the term use ie the doctors in-charge or the in-charge doctors?

Thank you for the suggestion. We have standardised the term using ‘doctors in charge’ throughout the paper.

p value less than (to use symbol <)

Thank you. Revision done as suggested. (Page 7, line 5)

The factors with p value <0.25 in simple logistic regression analysis were included in the multivariate model. [12]

Reviewer #2: Overall it's a very good study and very relevant to primary care. As a pioneer study, i think subsequent study can explore the reason for low prescription of MHT in relation to patients perception towards menopause symptoms and MHT. 

Thank you for the suggestion. We would consider this in the subsequent study.

I just want clarification regarding the questionnaire used. in the manuscript its written adapted from previous study and given the reference number (11). But in the references, reference number 11 is about 'multiple logistic regression. ( 11. Jr D, Lemeshow S, Sturdivant R. The Multiple Logistic Regression Model. In 2013. p. 35–47. ) I can't find the reference to the questionnaire used in this study. 

Thank you for informing this error. We have rechecked the references and correction done accordingly. 

The revised reference for the questionnaire is still number 11 (Yeganeh L, Boyle J, Teede H, Vincent A. Knowledge and attitudes of health professionals regarding menopausal hormone therapies. Climacteric. 2017 Aug;20(4):348–55.)

The earlier reference number 11, and subsequent reference has been revised accordingly (page 25 to 27)

And what step they do... is it only content validation? what language the original questionnaire used. Please provide me the original one and the method you use to validate the questionnaire.

We have added more details for clarity. regarding the process in validating this questionnaire. We did not proceed with exploratory/confirmatory factor analysis or reliability testing as the questionnaire items were analyzed individually as categorical data, and no composite scoring was used. (Page 6, line 1 to line 10)

'The tool used in this study is a questionnaire originally developed by previous researchers to assess health professionals' attitudes regarding menopause hormone therapy in Australia.[11] The questionnaire is in English and permission was obtained from the original authors to use it in our study. To better suit our local context, we made some adaptations to the original questionnaire. The questionnaire items were analyzed individually as categorical data, and no composite scoring was used. Content validation of the adapted questionnaire was performed by three senior family medicine specialists. Additionally, a pilot study was conducted involving 30 health professionals, who were excluded from the final sample. Feedback from the pilot study led to minor corrections, primarily in grammar and wording, to enhance clarity and ensure the questions were clearly understood by the respondents.'

For the original questionnaire, we obtained the permission from the author to be use in the study, however we did not have permission to distribute it to other party.

The original article can be found in DOI: 10.1080/13697137.2017.1304906

The questionnaire items were analysed individually as categorical data, and no composite scoring was used.

---

## [Decision Letter · Decision Letter 1]

11 Sep 2024

Prevalence of offering menopause hormone therapy among primary care doctors and its associated factors: A cross-sectional study

PONE-D-24-25801R1

Dear Dr. Ai Theng Cheong,

We’re pleased to inform you that your manuscript has been judged scientifically suitable for publication and will be formally accepted for publication once it meets all outstanding technical requirements.

Kind regards,

Ghanshyam G. Tejani

Academic Editor

PLOS ONE

Additional Editor Comments (optional):

This paper can be accepted in its current version.

Reviewers' comments:

Reviewer's Responses to Questions

**Comments to the Author**

1. If the authors have adequately addressed your comments raised in a previous round of review and you feel that this manuscript is now acceptable for publication, you may indicate that here to bypass the “Comments to the Author” section, enter your conflict of interest statement in the “Confidential to Editor” section, and submit your "Accept" recommendation.

Reviewer #1: All comments have been addressed

Reviewer #2: All comments have been addressed

2. Is the manuscript technically sound, and do the data support the conclusions?

Reviewer #1: Yes

Reviewer #2: Yes

3. Has the statistical analysis been performed appropriately and rigorously? 

Reviewer #1: Yes

Reviewer #2: Yes

4. Have the authors made all data underlying the findings in their manuscript fully available?

Reviewer #1: Yes

Reviewer #2: Yes

5. Is the manuscript presented in an intelligible fashion and written in standard English?

Reviewer #1: Yes

Reviewer #2: Yes

6. Review Comments to the Author

Reviewer #1: Thank you for rectifying and answering the queries. I am fine with the revision. I have no further comments.

Reviewer #2: All comments has been addressed. The Correction has been done to explain the questionnaire validation and refences also has been corrected.

7. PLOS authors have the option to publish the peer review history of their article (what does this mean?). If published, this will include your full peer review and any attached files.

Reviewer #1: **Yes: **Ping Foo Wong

Reviewer #2: No

---

## [Editor Report · Acceptance letter]

16 Sep 2024

PONE-D-24-25801R1 

PLOS ONE

Dear Dr. Cheong, 

I'm pleased to inform you that your manuscript has been deemed suitable for publication in PLOS ONE. Congratulations! Your manuscript is now being handed over to our production team.

Kind regards, 

on behalf of

Dr. Ghanshyam G. Tejani 

Academic Editor

PLOS ONE